# Recognizing and overcoming the greedy nature of learning in multi-modal deep neural networks

## Abstract

We hypothesize that due to the greedy nature of learning in multi-modal deep neural networks, these models tend to rely on just one modality while under-utilizing the other modalities. We observe empirically that such behavior hurts their overall generalization. To estimate the model's modality-wise dependency, we compute the gain on the accuracy when the model has access to an additional modality and refer to this gain as the *conditional utilization rate* of the added modality. In the experiments, we consistently observe an imbalance in conditional utilization rates between modalities, across multiple tasks and architectures. Since conditional utilization rate cannot be computed efficiently during training, we introduce an proxy for it based on the pace at which the model learns from each modality, which we refer to as *conditional learning speed*. We propose an algorithm to balance the conditional learning speed between modalities during training and demonstrate that it indeed addresses the issue of greedy learning. The proposed algorithm improves the model's generalization on three datasets: Colored MNIST (Kim et al., 2019), Princeton ModelNet40 (Wu et al., 2015), and NVIDIA Dynamic Hand Gesture Dataset (Molchanov et al., 2016).

## 1 Introduction

In real-world problems, each instance frequently has multiple modalities associated with it. For example, we detect cancer in both X-ray and ultrasound images. We seek clues from images to answer questions given in text. We are naturally interested in training deep neural networks (DNNs) end-to-end to learn from all available input modalities. We refer to such a training regime as a *multi-modal learning process* and DNNs resulting from this process as *multi-modal DNNs*.

Several recent studies have reported unsatisfactory performance of multi-modal DNNs in various tasks (Wang et al., 2020a; Wu et al., 2020; Gat et al., 2020; Cadene et al., 2019; Agrawal et al., 2016; Hessel & Lee, 2020; Han et al., 2021). For example, in Visual Question Answering, multi-modal DNNs were found to ignore the visual modality (presented as an image) and exploit statistical regularities shared between the textual modality (presented as a question) and the answer alone, resulting in poor generalization (Cadene et al., 2019; Gat et al., 2020; Agrawal et al., 2016). Similarly, in Human Action Recognition, multi-modal DNNs trained on images and audio were observed to under-perform uni-modal DNNs trained on images only (Wang et al., 2020a).

These earlier negative findings compel us to ask, *what prevents multi-modal DNNs from achieving better performance?* In order to answer this question, we first diagnose these DNNs as lacking utilization of all modalities by analyzing their conditional utilization rates. For a multi-modal DNN trained with two modalities, $m_0$ and $m_1$, the conditional utilization rate of $m_1$ given $m_0$, denoted by $\boldsymbol{u}(m_1|m_0)$, measures how important it is to use $m_1$, given the presence of $m_0$. It is computed as the relative difference in accuracy between two derived models from the DNN, one using both modalities and the other using only one modality. In several multi-modal learning tasks, we consistently observe a significant imbalance in conditional utilization rates between modalities. For example, we observe $\boldsymbol{u}(\text{RGB}|\text{depth}) = 0.01$ and $\boldsymbol{u}(\text{depth}|\text{RGB}) = 0.63$ for a DNN trained to identify gestures in videos using the NVIDIA Dynamic Hand Gesture Dataset (NVGesture) (Molchanov et al., 2016). It indicates that the DNN relies on the depth modality to make predictions and does not pay attention to the RGB

modality. These observations lead to a conjecture that the multi-modal learning process often results in models that under-utilize some of the input modalities.

As the focal point of this work, we put forward the *greedy learner hypothesis*. The greedy learner hypothesis states that a multi-modal DNN learns to rely on one of the input modalities, based on which it could learn faster, and does not continue to learn to use the other modalities. This greediness leads to an imbalance in conditional utilization rates between modalities. In other words, it prevents a multi-modal DNN from learning to exploit all available modalities and often results in worse generalization. It explains the challenge in training multi-modal DNNs and motivates us to design a better multi-modal learning algorithm.

According to the greedy learner hypothesis, it is the diverged speed at which a multi-modal DNN learns from different modalities that leads to an imbalance in conditional utilization rate. If we intervene in the training process to adjust these speeds, we may be able to prevent the hurtful imbalance across input modalities. We analyze the learning dynamics of model components and propose a metric named by conditional learning speed using the gradient norm and weight norm of models' parameters. It measures the relative learning speed at which the model learns from one modality against the other modality. We empirically show that it is a reasonable proxy for conditional utilization rate. We introduce a training algorithm, *balanced multi-modal learning* which guides the model to learn intentionally from one of the modalities according to the conditional learning speeds. We show that models trained with this algorithm learn to use all modalities appropriately and achieve stronger generalization on three multi-modal datasets: Colored MNIST dataset (Kim et al., 2019), ModelNet40 dataset of 3D objects (Su et al., 2015) and NVGesture Dataset (Molchanov et al., 2015).

## 2 PROBLEM SETUP

We consider two input modalities, referred to as $m_0$ and $m_1$, without loss of generality towards tasks with more than two modalities. We denote a multi-modal dataset by $\mathcal{D} = \left\{(\boldsymbol{x}^i, y^i)\right\}_{i=1}^{N}$, where $\boldsymbol{x} = (\boldsymbol{x}_{m_0}, \boldsymbol{x}_{m_1})$. We partition the dataset $\mathcal{D}$ into training, validation and test sets, denoted by $\mathcal{D}^{\text{train}}$, $\mathcal{D}^{\text{val}}$, and $\mathcal{D}^{\text{test}}$, respectively. The goal is then to use this data set to train a multi-modal DNN such that it accurately predicts $y$ from $\boldsymbol{x}$.

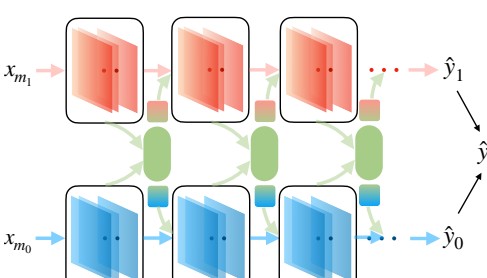

Figure 1: The multi-modal DNN with intermediate fusion that we study in this work. Blue and red blocks represent the two uni-modal networks learning from modalities $m_0$ and $m_1$. The fusion layers are marked with green, green-blue, and green-red. The predictions from the two uni-modal branches are denoted by $\hat{y}_0$ and $\hat{y}_1$. We denote the average of the two by $\hat{y}$ which is the model's prediction.

**Multi-modal DNNs**  We use a multi-modal DNN that has two uni-modal branches, each taking one modality as input. The two uni-modal branches are interconnected by layer-wise fusion modules. These fusion modules enable information flow from one branch to another. According to categorization of multi-modal fusion strategies in the deep learning literature, this is intermediate fusion (Ngiam et al., 2011; Atrey et al., 2010; Baltrušaitis et al., 2018). It has demonstrated competitive performance against multi-modal DNNs with late fusion in many tasks (Perez et al., 2018; Joze et al., 2020; Anderson et al., 2018; Wang et al., 2020b).

We implement every fusion module by a multi-modal transfer module (MMTM) (Joze et al., 2020). Each MMTM connects two layers from the two uni-modal branches. There is first the global average pooling applied over spatial dimensions to transform feature maps into a vector. We concatenate the two vectors and apply linear transformation. We refer to its output as context representation. Next, for each uni-modal branch, we implement a fully connected layer on the context representation and get a vector with a dimension of the number of feature maps. Feature maps are re-scaled by this vector before passing to the next layer of the uni-modal branch.

We train a multi-modal DNN $f$ on $\mathcal{D}^{\text{train}}$. Let $\hat{y}$ be the prediction of $f$ for an input $\boldsymbol{x}$:

$$\hat{y} = f(\boldsymbol{x}_{m_0}, \boldsymbol{x}_{m_1}).$$

As shown in Figure 1, $\hat{y} = \frac{1}{2}(\hat{y}_0 + \hat{y}_1)$, where $\hat{y}_0$ and $\hat{y}_1$ are the outputs of the two uni-modal branches. During training, the parameters of the multi-modal DNN are updated by stochastic gradient descent (SGD) to minimize the loss: $L = \text{CE}(y, \hat{y}_0) + \text{CE}(y, \hat{y}_1)$, where CE stands for cross-entropy. We refer to each of the cross-entropy losses as a modality-specific loss. We train the model until $\hat{y} = y$ for all samples in $\mathcal{D}^{\text{train}}$ and take the checkpoint of it when $\hat{y}$ reaches the highest accuracy on $\mathcal{D}^{\text{val}}$.

During training, each uni-modal branch largely focuses on its associate input modality. The fusion modules generate context representation using all modalities and feed such information to the uni-modal branches. Both $\hat{y}_0$ and $\hat{y}_1$ depend on information from both modalities. We end up with two functions, $f_0$ and $f_1$, corresponding to the two uni-modal branches:

$$\hat{y}_0 = f_0(\boldsymbol{x}_{m_0}, \boldsymbol{x}_{m_1}), \quad \hat{y}_1 = f_1(\boldsymbol{x}_{m_0}, \boldsymbol{x}_{m_1}).$$

To evaluate the multi-modal DNN's ability in using a single modality, we disable information sharing between branches ad derive $f_0'$ and $f_1'$, each taking only one of the input modalities as input. We describe how we derive them in detail in Appendix A.1. We use $\hat{y}_0'$ and $\hat{y}_1'$ to refer to their outputs:

$$\hat{y}_0' = f_0'(\boldsymbol{x}_{m_0}), \quad \hat{y}_1' = f_1'(\boldsymbol{x}_{m_1}).$$

In summary, we derive four models from $f$ and we define the ability of each model as its accuracy $A$ on $\mathcal{D}^{\text{test}}$. We group the four accuracies into two pairs: $(A(f_0), A(f_0'))$, and $(A(f_1), A(f_1'))$ and define the conditional utilization rates as bellow:

**Definition 2.1 (Conditional utilization rate)** *For a multi-modal DNN, $f$, taking two modalities $m_0$ and $m_1$ as inputs, its conditional utilization rates for $m_1$ and $m_0$ are defined as $\boldsymbol{u}(m_1|m_0) = \frac{A(f_0) - A(f_0')}{A(f_0)}$ and $\boldsymbol{u}(m_0|m_1) = \frac{A(f_1) - A(f_1')}{A(f_1)}$, respectively.*

The conditional utilization rate is the relative change in accuracy between the two models within each pair. The conditional utilization rate of $m_0$, $u(m_0|m_1)$, measures how important it is for $f$ to use $m_0$ together with $m_1$. In other words, it estimates the marginal contribution that $m_0$ has in increasing the accuracy of $\hat{y}_1$ depending on $m_1$.

**A multi-modal learning process** Given a DNN's architecture and a dataset, every time we run a multi-modal learning process, we perform the following steps: (1) sample a value of learning rate randomly from a given range, (2) initialize the DNN's parameters randomly, (3) update them to fit $\mathcal{D}^{\text{train}}$ with SGD, and (4) select the model checkpoint with the highest accuracy on $\mathcal{D}^{\text{val}}$.

## 3 THE GREEDY LEARNER HYPOTHESIS

In this section, we introduce the greedy learner hypothesis to explain challenges observed in training multi-modal DNNs. Before describing our hypothesis, we discuss some assumptions on the multi-modal data and preliminary observations made in the multi-modal learning literature.

### 3.1 ASSUMPTIONS IN MULTI-MODAL LEARNING

First, in multi-modal learning literature (Blum & Mitchell, 1998; Sridharan & Kakade, 2008), both modalities are assumed to be predictive of the target, i.e., $I(\boldsymbol{Y}, \boldsymbol{X}_{m_0}) > 0$ and $I(\boldsymbol{Y}, \boldsymbol{X}_{m_1}) > 0$, where $I$ denotes the mutual information. In order to minimize one of the modality-specific losses, e.g. $\text{CE}(y, \hat{y}_0)$, one can either update the parameters of the fusion layers that pass information from $m_1$ to $\hat{y}_0$, or the parameters of the uni-modal branch taking $m_0$ as input, or both.

Second, modalities are predictive of the target at different degrees. For example, it has been observed that when training DNNs on each modality separately, they usually do not reach the same performance (Wang et al., 2020b; Joze et al., 2020). In addition, multi-modal DNNs exhibit varying performance when being trained on different subsets of modalities present for the task (Weng et al., 2021; Pérez-Rúa et al., 2019; Liu et al., 2018).

Third, modalities are learned by the model at different speeds. This has been observed in both uni-modal DNNs and Multi-modal DNNs (Wang et al., 2020a; Wu et al., 2020).

### 3.2 MULTI-MODAL LEARNING PROCESS IS GREEDY

Before formalizing our greedy learner hypothesis that explains the behaviour of multi-modal DNNs, we first demonstrate that we can tell whether the model utilizes all modalities by analyzing its conditional utilization rates.

Given a multi-modal DNN, $f$, with two input modalities, $m_0$ and $m_1$, we compute the difference between conditional utilization rates, $\boldsymbol{u}(m_1|m_0)$ and $\boldsymbol{u}(m_0|m_1)$ as

$$\mathrm{d}_{\mathrm{util}}(f) = \boldsymbol{u}(m_1|m_0) - \boldsymbol{u}(m_0|m_1).$$

If we observe high $|\mathrm{d}_{\mathrm{util}}(f)|$, we say that $f$ exhibits imbalance in utilization between modalities.

The difference between conditional utilization rates is bounded between $-1$ and $1$. When it is close to $-1$ or $1$, the model benefits only from one of the modalities given the other but not vice versa. This implies that the model's ability to predict $\hat{y}_0$ and $\hat{y}_1$ comes only from one of the modalities.

We now introduce the *greedy learner hypothesis*:

> We call a multi-modal learning process *greedy*, when it trains models to rely on only one of the available modalities. A greedy multi-modal learning process cannot avoid producing models that exhibit a high degree of imbalance in utilization between modalities. We *hypothesize* that a multi-modal learning process, in which a multi-modal DNN is trained to minimize the sum of modality-specific losses, is greedy.

Let $\mathbb{E}[\widehat{\mathrm{d}}_{\mathrm{util}}]$ denote the expectation of $\widehat{\mathrm{d}}_{\mathrm{util}}$ over the empirical distribution of models resulting from a multi-modal learning process (as defined in §2). The absolute value of $\mathbb{E}[\widehat{\mathrm{d}}_{\mathrm{util}}]$ is associated with the greediness of the process. The higher the $|\mathbb{E}[\widehat{\mathrm{d}}_{\mathrm{util}}]|$ is, the greedier the learning process is.

We propose the following *conjectures* consistent with the greedy learner hypothesis:

1. There exists an $\epsilon > 0$, s.t. $|\mathbb{E}[\widehat{\mathrm{d}}_{\mathrm{util}}]| > \epsilon$, as long the multi-modal dataset has the last two properties from §3.1. Otherwise, $\widehat{\mathrm{d}}_{\mathrm{util}}$ is distributed symmetrically around zero and $\mathbb{E}[\widehat{\mathrm{d}}_{\mathrm{util}}] = 0$.

2. The stronger the regularization we apply to the DNNs' parameters in training, the higher the $|\mathbb{E}[\widehat{\mathrm{d}}_{\mathrm{util}}]|$ is.

We expect the first scenario in the first conjecture to be true for all real-world multi-modal tasks. To test the second scenario in the first conjecture, we construct datasets of two identical modalities.

We suspect that the implicit regularization of DNNs is a factor impacting the greediness of learning in multi-modal DNNs. The implicit regularization of DNNs commonly refers to a consensus that DNNs have the tendency to lean towards simplicity during training and it explains their strong generalization (Valle-Perez et al., 2019; Zhang et al., 2017; 2021; Smith et al., 2021). Since it is difficult to precisely measure and control such implicit regularization, alternatively, we construct the second conjecture using explicit regularization applied to the DNNs' parameters.

In order to validate the above hypothesis and conjectures empirically, we conduct experiments on several multi-modal datasets using different network architectures (cf. §5.1). We present our observations in §5.2 and §5.3. We show that the multi-modal learning process cannot avoid producing models that exhibit a high degree of imbalance in utilization between modalities. Further, in the next section, we propose a metric for measuring the speeds at which the model learns from different modalities and we show the imbalance in such speeds is correlated with the imbalance in utilization.

## 4 MAKING MULTI-MODAL LEARNING LESS GREEDY

We aim to make multi-modal learning less greedy by controlling the speed at which a multi-modal DNN learns to rely on each modality. To this end, we define conditional learning speed to measure the speed at which the DNN learns from one modality. It serves as an efficient proxy to the conditional utilization rate of the corresponding modality, as shown empirically in §5.2 and §5.3. We then propose the balanced multi-modal learning algorithm, which controls the difference in conditional learning speed between modalities that the model exhibits during training.

## 4.1 Conditional learning speed

As demonstrated in §3.2, the imbalance in conditional utilization rates is a sign of the model exploiting the connection between the target and only one of the input modalities, ignoring cross-modal information. However, conditional utilization rates are measured after training is done, making it expensive to use them in real-time during training. We instead derive a proxy metric, called *conditional learning speed*, that captures relative learning speed between modalities during training.

Let us revisit the architecture of the multi-modal DNN. We denote the parameters of each uni-modal branch taking $m_0$ and $m_1$ as inputs, by $\boldsymbol{\theta}_0$ and $\boldsymbol{\theta}_1$. Layers of the fusion modules marked with green and green/blue in Figure 1, are part of the function mapping $\boldsymbol{x}$ to $\hat{y}_0$. Let $\boldsymbol{\theta}'_0$ refer to their parameters. Analogously, the layers marked with green and green/red in Figure 1 are part of the function mapping $\boldsymbol{x}$ to $\hat{y}_1$. We denote their parameters by $\boldsymbol{\theta}'_1$. In this way, we divide the model's parameters into two pairs: $(\boldsymbol{\theta}_0, \boldsymbol{\theta}'_0)$ and $(\boldsymbol{\theta}_1, \boldsymbol{\theta}'_1)$.[1]

For parameters $\boldsymbol{\theta}$, let $\boldsymbol{\theta}_{(i-1)}$ and $\boldsymbol{\theta}_{(i)}$ denote its value before and after the gradient descent step $i$, and $\boldsymbol{G} = \frac{\partial L}{\partial \boldsymbol{\theta}}|_{\boldsymbol{\theta}_{(i-1)}}$. We define the model's conditional learning speed as:

**Definition 4.1 (Conditional learning speed)** *Given a multi-modal DNN, $f$, with two input modalities, $m_0$ and $m_1$, the conditional learning speeds of these modalities after $t$ training steps, are*

$$\boldsymbol{s}(m_1|m_0; t) = \log \frac{\sum_{i=1}^{t} \mu(\boldsymbol{\theta}'_0; i)}{\sum_{i=1}^{t} \mu(\boldsymbol{\theta}_0; i)}, \quad \boldsymbol{s}(m_0|m_1; t) = \log \frac{\sum_{i=1}^{t} \mu(\boldsymbol{\theta}'_1; i)}{\sum_{i=1}^{t} \mu(\boldsymbol{\theta}_1; i)},$$

*where $\mu(\boldsymbol{\theta}; i) = ||\boldsymbol{G}||_2^2/||\boldsymbol{\theta}_{(i)}||_2^2$ quantifies the change of $f$ that comes from updating $\boldsymbol{\theta}$ at the $i^{th}$ step, which can be also interpreted as the effective update on $\boldsymbol{\theta}$.*

This definition of $\mu(\boldsymbol{\theta}; i)$ is inspired by discussion on the effective update of parameters in the literature (Van Laarhoven, 2017; Zhang et al., 2019; Brock et al., 2021; Hoffer et al., 2018). When normalization techniques, such as batch normalization (Ioffe & Szegedy, 2015), are applied to the DNNs, the key property of the weight vector, $\boldsymbol{\theta}$, is its direction, i.e., $\boldsymbol{\theta}/||\boldsymbol{\theta}||_2^2$. Thus, we measure the update on $\boldsymbol{\theta}$ using the norm of its gradient normalized by its norm.

The conditional learning speed, $\boldsymbol{s}(m_1|m_0; t)$ (and analogously $\boldsymbol{s}(m_0|m_1; t)$), is the log-ratio between the learning speed of $\boldsymbol{\theta}'_0$ and that of $\boldsymbol{\theta}_0$. Because $\boldsymbol{\theta}'_0$ carries information from $m_1$ to $\hat{y}_0$ and information from $m_0$ to $\hat{y}_0$ is carried by $\boldsymbol{\theta}_0$, $\boldsymbol{s}(m_1|m_0; t)$ reflects how fast the model learns from $m_1$ relative to $m_0$, after the first $t$ steps.

We further compute the difference between $\boldsymbol{s}(m_1|m_0; t)$ and $\boldsymbol{s}(m_0|m_1; t)$ as:

$$\mathrm{d}_{\mathrm{speed}}(f; t) = \boldsymbol{s}(m_1|m_0; t) - \boldsymbol{s}(m_0|m_1; t),$$

analogous to $\mathrm{d}_{\mathrm{util}}(f)$. For each model, we report $\mathrm{d}_{\mathrm{speed}}(f; T)$ as $\mathrm{d}_{\mathrm{speed}}(f)$ where the model takes $T$ steps until reaching the highest accuracy on $\mathcal{D}^{\mathrm{val}}$. We anticipate that the conditional learning speed would serve as a proxy for the conditional utilization rate and we say $\mathrm{d}_{\mathrm{speed}}(f; t)$ would predict $\mathrm{d}_{\mathrm{util}}(f)$ at the end of training. In §5.2 and §5.3, we show the distributions of $\widehat{\mathrm{d}}_{\mathrm{speed}}$ and $\widehat{\mathrm{d}}_{\mathrm{util}}$ side-by-side to validate this.

## 4.2 Balanced multi-modal learning

Because $\mathrm{d}_{\mathrm{speed}}(f, t)$ is predictive of the imbalanced utilization between modalities, we can take advantage of $\mathrm{d}_{\mathrm{speed}}(f, t)$ to balance conditional utilization on-the-fly. In addition to training the network normally with both modalities, we accelerate the model to learn from either modality alternately to balance the conditional learning speeds of them. See Algorithm 1 for an overall description of the algorithm.

We refer to the training steps at which we perform forward and backward passes normally as *regular steps*. We introduce *re-balancing steps* at which we update one of the uni-modal branches intentionally in order to accelerate the model to learn from the corresponding modality. See Appendix A.2 for the full explanation of the re-balancing step.

---

[1]Some parameters are both in $\boldsymbol{\theta}'_0$ and $\boldsymbol{\theta}'_1$.

To warm-up the model, we perform only regular steps in the first training epoch. Then we switch from regular steps to re-balancing steps if $|\mathrm{d}_{\mathrm{speed}}(t)| > \alpha$, where $\alpha$ is a hyperparameter, referred to as the *imbalance tolerance parameter*. The training takes $Q$ re-balancing steps before returning to regular mode. We refer to the hyperparameter $Q$ as the *re-balancing window size*.

---

**Algorithm 1:** Balanced multi-modal learning

---

**Input:** $Q$, re-balancing window size;
      $\alpha$, imbalance tolerance parameter
      $T$, # of training steps;
      $T_1$, # of steps in the $1^{th}$ training epoch

Let M denote the accumulated effective update ($\mu$) in regular steps
$M_{\boldsymbol{\theta}_0}, M_{\boldsymbol{\theta}_0'} = 0, 0$;
$M_{\boldsymbol{\theta}_1}, M_{\boldsymbol{\theta}_1'} = 0, 0$;
**for** $t \leftarrow 1$ **to** $T_1$ **do**
    Take a regular step;
    Update $M_{\boldsymbol{\theta}_0}, M_{\boldsymbol{\theta}_0'}, M_{\boldsymbol{\theta}_1}, M_{\boldsymbol{\theta}_1'}$
**end**
regular_mode = True

**for** $t \leftarrow T_1$ **to** $T$ **do**
    **if** *regular_mode* **then**
        Take a regular step;
        Update $M_{\boldsymbol{\theta}_0}, M_{\boldsymbol{\theta}_0'}, M_{\boldsymbol{\theta}_1}, M_{\boldsymbol{\theta}_1'}$;
        Compute $\mathrm{d}_{\mathrm{speed}}$;
        **if** $|\mathrm{d}_{\mathrm{speed}}| > \alpha$ **then**
            $q = 0$; regular_mode = False
        **end**
    **else**
        $q \leftarrow q + 1$;
        **if** $q = Q$ **then** regular_mode = True;
        Take a re-balancing step to accelerate learning from $m_0$ **if** $\mathrm{d}_{\mathrm{speed}} > 0$ **else** $m_1$;
    **end**
**end**

---

## 5 EXPERIMENTS AND RESULTS

### 5.1 DATASETS, TASKS AND BASELINES

**Colored-and-gray-MNIST** (Kim et al., 2019) is a synthetic dataset based on MNIST (LeCun et al., 1998). In the training set of 60,000 examples, each example has two images, a gray-scale image and a monochromatic image, with color strongly correlated with its digit label. For the validation set of 10,000 examples, each example also has a gray-scale image and a corresponding monochromatic image however with a low correlation between the color and its label. We consider the monochromatic image as the first modality $m_0$ and the gray-scale one as the second modality $m_1$. We use a neural network with four convolutional layers as the uni-modal branch and employ three MMTMs to connect them. The corresponding uni-modal DNNs trained on the monochromatic images and the gray-scale images achieve the validation accuracies of 43% and 98%, respectively. We use this synthetic dataset mainly to demonstrate the proposed greedy learner hypothesis.

**ModelNet40** is one of the Princeton ModelNet datasets (Wu et al., 2015) with 3D objects of 40 categories (9,483 training samples and 2,468 test samples). We use the task that is to classify a 3D object based on the 2D views of its front and back (Su et al., 2015). Each example is a 3D object rendered as 2D images of 224×224 pixels. For the uni-modal branches, we use ResNet18 (He et al., 2016) and apply MMTMs in the three final residual blocks. The uni-modal DNNs achieve 91.6% and 93.1% accuracy, when learning from the front view ($m_0$) and the rear view ($m_1$), respectively.

NVIDIA Dynamic Hand Gesture Dataset (or **NVGesture** (Molchanov et al., 2015)), consists of 1,532 video clips (1,050 training and 482 test ones) of hand gestures in 25 classes. We sample 20% training examples as the validation set and use depth and RGB as the two modalities. We adopt the data preparation steps used in Joze et al. (2020) and use the I3D architecture (Carreira & Zisserman, 2017) as uni-modal branches and MMTMs as fusion modules in the six final inception modules.

We provide examples of each dataset and details on data preprocessing in § A.4 in Appendix.

### 5.2 VALIDATING THE GREEDY LEARNER HYPOTHESIS

In this section, we run the conventional multi-modal learning process on seven tasks with different input and output pairs to validate the first conjecture from §3.2 experimentally and the proposed greedy learner hypothesis.

**Study design** For each task introduced in §5.1, in addition to the original dataset, we construct a dataset with two identical input modalities by copying one of the modalities. For example, when using the colored-and-gray-MNIST dataset, we predict the digit class using two identical gray-scale images. We train a multi-modal DNN on these dataset as explained below for each task:

- Colored-and-gray-MNIST: we train multi-modal DNNs using SGD with the momentum coefficient of 0.9 and a batch size of 128. We sample 20 learning rates at random from the interval $[10^{-5}, 1]$ on a logarithmic scale. We train the model four times using each of the learning rate and random initialization of the parameters. In total, we train 80 models.

- ModelNet40: we use SGD without momentum and use minibatches of eight examples. We select nine learning rates from $10^{-3}$ to 1 and train model using each learning rate for three times. This ends up with 27 models.

- NVGesture: we use a batch size of four, SGD with momentum of 0.9, and uniformly sample 20 learning rates from the interval $[10^{-4}, 10^{-1.5}]$ on a logarithmic scale. We train the model three times using each learning rate, resulting in 60 models in total.

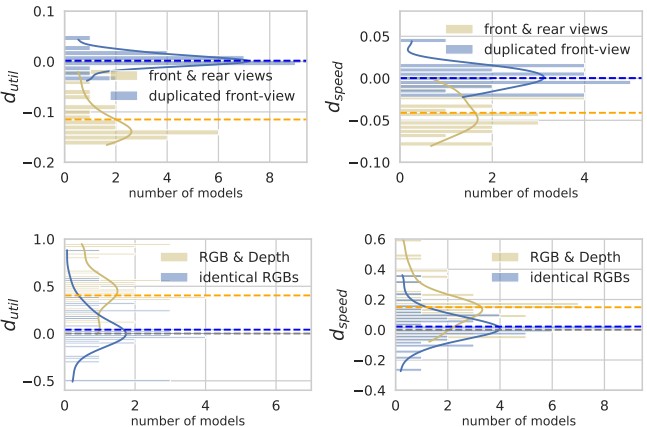

Figure 2: Histograms and estimated density functions of $\widehat{d}_{util}$ and $\widehat{d}_{speed}$ of models trained using ModelNet40 (top) or NVGesture (bottom). We mark zero and $\mathbb{E}[\widehat{d}_{util}]$ with dashed lines. Many models have high $\widehat{d}_{util}$. We see $\widehat{d}_{util}$ is distributed asymmetrically around zero when using two different input modalities. When using two identical input modalities, it is distributed symmetrically around zero. We see $\widehat{d}_{speed}$ exhibits similar distributions as $\widehat{d}_{util}$.

**Results** We present the results for the experiments on ModelNet40 and NVGesture in Figure 2 and on Colored-and-gray-MNIST in Figure 2 in Appendix.

First, many models have high $|d_{util}|$. This confirms that the multi-modal learning process encourages the model to rely on one modality and ignore the other one, which is consistent with our hypothesis. We make this observation across all tasks, confirming that the conventional multi-modal learning process is greedy regardless of network architectures and tasks.

Second, $\widehat{d}_{util}$ is distributed symmetrically around zero, and $\mathbb{E}[\widehat{d}_{util}]$ is approximately 0.0, for all the experiments using two identical input modalities. On the other hand, if we use two distinct modalities, $\widehat{d}_{util}$ is distributed asymmetrically, and we observe $|\mathbb{E}[\widehat{d}_{util}]|$ of approximately 0.3, 0.1 and 0.4 for colored-and-gray-MNIST, ModelNet40 and NVGesture, respectively. This validates the first conjecture in §3.2.

Third, by observing conditional learning speed $\widehat{d}_{speed}$, we can draw the same conclusions. In fact, the distributions of $\widehat{d}_{speed}$ largely replicate the distributions of $\widehat{d}_{util}$. It validates our greedy learner hypothesis which blames the varying rate at which the learner learns to rely on each modality. It moreover confirms $d_{speed}$ is an appropriate proxy to use to re-balance multi-modal learning.

We analyze the model's behavior when using different learning rates. Interestingly, we see relatively balanced conditional utilization rates when using high learning rates, as shown in Figure 3 in Appendix. This observation indicates that high learning rates implicitly calibrate the learning pace between modalities. We leave this for future analysis.

## 5.3 STRONG REGULARIZATION ENCOURAGES GREEDINESS

We investigate L1 regularization's impact on multi-modal DNNs and demonstrate that, as the second conjecture in §3.2 says, strong regularization encourages greediness in multi-modal learning.

**Study design**   We apply L1 regularization to the multi-modal DNNs. That is, we train the networks to optimize the loss $L' = L + \lambda||\boldsymbol{\theta}||_1$, where $L$ is the classification loss in §2, $\boldsymbol{\theta}$ stands for all model parameters and $\lambda$ is the weight on the regularizer.

We measure the effect of $\lambda$ on the network $f$ by computing the fraction of its parameters smaller than $10^{-7}$. We denote this quantity by $R(f)$. Since L1 regularization encourages sparsity of the network's parameters, as shown in Appendix Figure 4, the larger the $\lambda$ we use, the higher the $R(f)$ we observe.

We conduct this study with ModelNet40, using the front and rear views. We choose ten values from an interval $[10^{-9}, 10^{-3}]$ as $\lambda$. We use SGD without momentum, set the learning rate to 0.1 and batch size to eight. Using each combination of hyperparameters, we repeat training for three times with random initialization and get three models.

**Results**   As shown in Figure 3, $|\,\mathrm{d_{util}}(f)|$ increases along $\lambda$, especially when $\log(\lambda) \geq -5$. We also see that $|\,\mathrm{d_{util}}(f)|$ is positively correlated with $R(f)$. In other words, the stronger the regularization is, the larger the imbalance in utilization between modalities we observe. We see that $|\,\mathrm{d_{speed}}|$ follows the same trend as $|\,\mathrm{d_{util}}|$. Again, it supports our choice of using the conditional learning speed to predict the conditional utilization rate.

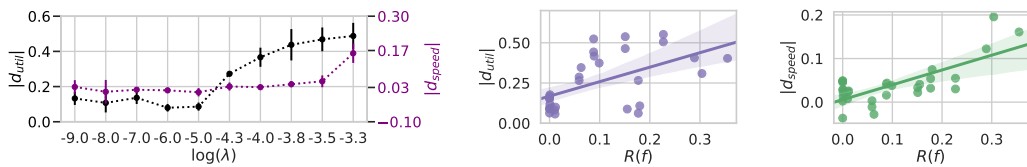

Figure 3: The observed $|\,\mathrm{d_{util}}|$ and $|\,\mathrm{d_{speed}}|$ for models trained with different weights ($\lambda$) on the L1 regularizer. *Left*: $|\,\mathrm{d_{util}}|$ and $|\,\mathrm{d_{speed}}|$ as a function of $\log(\lambda)$. *Middle*: $|\,\mathrm{d_{util}}|$ increases along $R(f)$. *Right*: The $|\,\mathrm{d_{speed}}|$ increases along $R(f)$. The multi-modal learning process exhibit higher greed if we introduce stronger regularization on the multi-modal DNNs.

## 5.4   BALANCED MULTI-MODAL LEARNING

Besides the proposed training algorithm (cf. §4.2, referred to as *guided*), we introduce a variant of it in Appendix A.3, referred to as *random*. This algorithm is motivated by Modality Dropout (Neverova et al., 2015) but better suited to the multi-modal DNNs with intermediate fusion. We consider it a stronger baseline that can also balance learning from inputs of different modalities.

**Calibrated modality utilization**   We train multi-modal DNNs as described in §5.2, using the guided, the random, and the conventional training algorithm (referred to as *vanilla*). For ModelNet40, we set the imbalance tolerance parameter $\alpha$ to 0.01 and the re-balancing window size $Q$ to 5. For NVGesture, we use $\alpha$ of 0.1 and $Q$ of 5.[2] Results are shown in Figure 4. Models trained with the guided algorithm have lower $|\widehat{\mathrm{d}}_{\mathrm{util}}|$ compared to the vanilla algorithm. On NVGesture, we obtain $\mathbb{E}(\widehat{\mathrm{d}}_{\mathrm{util}})$ of approximately 0.3 and 0.4 for models trained with the guided and the vanilla algorithm. On ModelNet40, we obtain $\mathbb{E}(\widehat{\mathrm{d}}_{\mathrm{util}})$ of approximately -0.0, and -0.1 for models trained with the guided and the vanilla algorithm.

The random algorithm also calibrates modality utilization effectively (see Figure 7 in Appendix). We will see in the next section that it helps less on generalization compared to the guided algorithm.

**Improved generalization performance**   We compare the generalization ability of multi-modal DNNs trained by the three algorithms (guided, random and vanilla) and the RUBi learning strategy (Cadene et al., 2019). For each algorithm, we train each model three times with the same learning rate. We use 0.01, 0.1 and 0.01 as learning rate for Colored-and-gray-MNIST, ModelNet40 and NVGesture respectively. We use $\alpha$ of 0.1 for Colored-and-gray-MNIST and NVGesture and 0.01 for ModelNet40. We set $Q$ of 5 for all three datasets. For NVGesture, we add one experiment where we initialize the model with parameters pre-trained using the Kinetics dataset (Carreira & Zisserman, 2017) in addition to random initialization. We refer to this setting as "NVGesture-pretrained" and to the other one as "NVGesture-scratch".

---

[2]We provide studies on the model's sensitivity to $Q$ and $\alpha$ in Figure 5 and Figure 6 in Appendix.

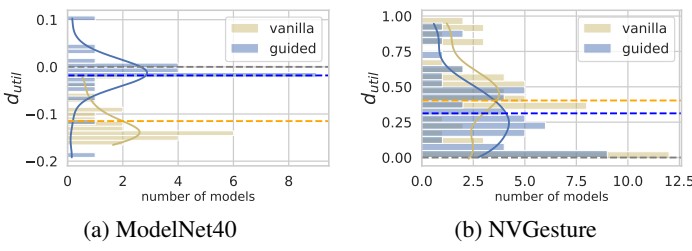

(a) ModelNet40     (b) NVGesture

Figure 4: Histograms and estimated density functions of $\widehat{d}_{\mathrm{util}}$ of models trained using the guided and the vanilla algorithm. We use dashed lines to mark zero and $\mathbb{E}[\widehat{d}_{\mathrm{util}}]$ over the set of models trained with each of the algorithm. The guided algorithm is less greedy than the vanilla one.

We report means and standard deviations of the models' test accuracies in Table 1.[3] RUBi does not show consistent improvement across tasks compared to the vanilla algorithm. The guided algorithm improves the models' generalization performance over all three other methods in all four cases.

Table 1: Test accuracy of models trained with the vanilla, random and guided algorithms.

|  | Colored-and-gray-MNIST | ModelNet40 | NVGesture-scratch | NVGesture-pretrained |
|---|---|---|---|---|
| **vanilla** | 45.26±0.46 | 90.09±0.58 | 79.81±1.14 | 83.20±0.21 |
| **RUBi** | 44.79±0.62 | 90.45±0.58 | 79.95±0.12 | 81.60±1.28 |
| **random** | 74.07±2.75 | 91.36±0.10 | 79.88±0.90 | 82.64±0.84 |
| **guided** | **91.01±1.20** | **91.37±0.28** | **80.22±0.73** | **83.82±1.45** |

## 6 RELATED WORK

Previous works on multi-modal learning focus more on architectural designs of the DNNs (Ngiam et al., 2011; Tran et al., 2015; Lazaridou et al., 2015; Wang et al., 2020b; Pérez-Rúa et al., 2019). Recently, unbalanced learning dynamics and modality-wise utilization in the end-to-end trained multi-modal classifiers were discussed (Goyal et al., 2017; Han et al., 2021; Wang et al., 2020a; Gat et al., 2020; Hessel & Lee, 2020; Collell & Moens, 2018; Winterbottom et al., 2020; Sun et al., 2021). For example, Wang et al. (2020b) design a parameter-free multi-modal fusion framework that dynamically exchanges channels between the uni-modal branches. Wang et al. (2020a) estimate the uni-modal branches' generalization and overfitting speeds and calibrate the learning through loss re-weighting. Their weight estimation slows down training and relies on a representative subset of the training data. Finally Gat et al. (2020) point out the bias in the multi-modal data as the cause of the issue. They propose to supply inputs with Gaussian perturbations to the model and regularize it by maximizing functional entropies. In this work, we explain this phenomenon by the greedy nature of learning in multi-modal DNNs and propose an algorithm based on the training dynamics of model. Our proposed method does not introduce any additional computation and only relies on the norm of the gradients and weights computed during training. It does not require architectural changes of the multi-modal DNNs and can be easily modified to accommodate different fusion modules. In addition to the multi-modal classifiers, the unbalanced modality-wise utilization has been identified in the multi-modal pre-trained models (Li et al., 2020; Cao et al., 2020). We expect that the insights we provide in this work are applicable to the self-supervised multi-modal training framework too.

## 7 CONCLUSIONS

Our work demonstrates that the end-to-end trained multi-modal DNNs rely on one of the input modalities to make predictions while leaving the other modalities underutilized. We hypothesize that it is due to the greedy nature of learning in multi-modal DNNs, and validated our statements experimentally on three multi-modal datasets. We illustrated that using the proposed algorithm to balance the models' learning from different modalities enhances generalization. This result emphasizes that the adequate modality utilization is a desired property that a model should achieve in multi-modal learning.

---

[3]Results on NVGesture are not directly comparable with numbers in other works since we use 20% training samples as the validation data.

## ETHICS STATEMENT

This work focuses on analyzing the challenges of training multi-modal deep neural networks. It does not involve any human subjects. We do not anticipate any potentially harmful consequences for society due to the publication of this work. As we use publicly available datasets in our study, we do not anticipate privacy and security issues. We claim no conflicts of interest.

## REPRODUCIBILITY STATEMENT

We provide complete descriptions of the used datasets and the applied processing steps in §5.1 and in Appendix §A.4. We state our experimental configurations for each study in §5.1, §5.2, and §5.3. We plan to publish the source code once the paper is released.

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

# A APPENDIX

## A.1 DERIVING $f'_0$ AND $f'_1$ FROM THE MULTI-MODAL DNNS

We denote the $k^{th}$ fusion module by $g_k$. For an input $\boldsymbol{x}$, let $h_k^{m_0}(\boldsymbol{x})$ and $h_k^{m_1}(\boldsymbol{x})$ denote the feature vectors from the layer in the uni-modal branches where $g_k$ is implemented.

To derive $f'_0(\boldsymbol{x}_{m_0})$ and $f'_1(\boldsymbol{x}_{m_1})$, we force each uni-modal branch to make predictions based on the single modality. Precisely, for each uni-modal branch, we aggregate its feature vectors over the whole training set and compute their average:

$$\overline{h}_k^{m_0} = \frac{1}{N} \sum_{\boldsymbol{x'}} h_k^{m_0}(\boldsymbol{x'}), \qquad \overline{h}_k^{m_1} = \frac{1}{N} \sum_{\boldsymbol{x'}} h_k^{m_1}(\boldsymbol{x'}).$$

Let $w_k^{m_0}(\boldsymbol{x})$ and $w_k^{m_1}(\boldsymbol{x})$ denote the context information $g_k$ gives:

$$w_k^{m_0}(\boldsymbol{x}) = g_k(h_k^{m_0}(\boldsymbol{x}), \overline{h}_k^{m_1}), \qquad w_k^{m_1}(\boldsymbol{x}) = g_k(\overline{h}_k^{m_0}, h_k^{m_1}(\boldsymbol{x})).$$

In this way, we cut off the cross-modal information sharing between the uni-modal branches while limiting the distribution shift for both the fusion modules and the layers in the uni-modal branches.

In our multi-modal DNN, we employ the fusion module as a Multi-Modal Transfer Module (MMTM). We adopt the annotation used by Joze et al. (2020). A MMTM takes two sets of tensors, $\boldsymbol{A} \in \mathbb{R}^{N_1 \times \cdots \times N_K \times C}$ and $\boldsymbol{B} \in \mathbb{R}^{M_1 \times \cdots \times M_L \times C'}$, each as feature maps of a convolutional layer of a uni-modal branch. Here, $N_i$ and $M_i$ represent the spatial dimensions of each feature map, and $C$ and $C'$ represent the number of feature maps.

In order to derive $f'_0(\boldsymbol{x}_{m_0})$ and $f'_1(\boldsymbol{x}_{m_1})$, we have operations in a MMTM written as:

$$\overline{S}_A(c) = \frac{1}{N_{training}} \sum_{i=1}^{N_{training}} S_A^i(c), \qquad \overline{S}_B(c) = \frac{1}{N_{training}} \sum_{i=1}^{N_{training}} S_B^i(c)$$

$$Z_A = \boldsymbol{W}[S_A, \overline{S}_B], \qquad\qquad Z_B = \boldsymbol{W}[\overline{S}_A, S_B]$$

$$E'_A = W_A Z_A + b_A, \qquad\qquad E'_B = W_B Z_B + b_B$$

$$\tilde{\boldsymbol{A}}' = 2 \times \sigma(E'_A) \odot \boldsymbol{A}, \qquad\qquad \tilde{\boldsymbol{B}}' = 2 \times \sigma(E'_B) \odot \boldsymbol{B},$$

where $[\cdot, \cdot]$ represents the concatenation operation; $W_i$ and $b_i$ are parameters of the fully connected layers; $\odot$ is the channel-wise product operation; and $\sigma(\cdot)$ is the sigmoid function.

## A.2 THE RE-BALANCING STEP

Suppose that we have $K$ fusion modules and let $w_k^{m_0}(\boldsymbol{x})$ and $w_k^{m_1}(\boldsymbol{x})$ denote the output of the $k^{th}$ fusion module. At the $t$ training step in the balanced multi-modal learning process, we have

$$\overline{w}_k^{m_0} = \frac{1}{N_t} \sum_{\boldsymbol{x'} \in \mathcal{D}_{\text{passed}}^t} w_k^{m_0}(\boldsymbol{x'}), \qquad \overline{w}_k^{m_1} = \frac{1}{N_t} \sum_{\boldsymbol{x'} \in \mathcal{D}_{\text{passed}}^t} w_k^{m_1}(\boldsymbol{x'}),$$

where $\mathcal{D}_{\text{passed}}^t$ denotes all samples appeared in the previous regular training steps until now. For example, in a re-balancing step on $m_0$, we aim to accelerate the model's learning from $m_0$, and we have

$$\hat{y}_0 = \phi_0(\boldsymbol{x}_{m_0}, \overline{w}_1^{m_0}, \cdots, \overline{w}_K^{m_0}), \quad \hat{y}_1 = \phi_1(\boldsymbol{x}_{m_1}, w_1^{m_1}(\boldsymbol{x}), \cdots, w_K^{m_1}(\boldsymbol{x})),$$

where $\boldsymbol{x} = (\boldsymbol{x}_{m_0}, \boldsymbol{x}_{m_1})$ is the input at this step, and $\phi_0$ and $\phi_1$ denote the uni-modal branch take $m_0$ and $m_1$ as inputs.

## A.3 RANDOM VERSION OF THE BALANCED MULTI-MODAL LEARNING ALGORITHM

We present a *random* version of the balanced multi-modal learning algorithm in Algorithm 1. Similar to the guided version, we perform regular steps in the first epoch. Afterwards, at each time, we let the model take the step that is randomly sampled from: regular step, re-balancing step on $m_0$, and re-balancing step on $m_1$.

---

**Algorithm 1:** Random version of the balanced multi-modal learning

| | |
|---|---|
| **Input:** $T$, # of training steps; 
 $\quad T_1$, # of steps in the $1^{th}$ training epoch; 
 $\quad \Omega_{step} = \{$ regular step, 
 $\qquad\qquad$ re-balancing step to accelerate 
 $\qquad\qquad$ learning from $m_0$, 
 $\qquad\qquad$ re-balancing step to accelerate 
 $\qquad\qquad$ learning from $m_1 \}$ | **for** $t \leftarrow 1$ **to** $T_1$ **do** 
 $\quad\mid$ Take a regular step; 
 **end** 
 **for** $t \leftarrow T_1$ **to** $T$ **do** 
 $\quad\mid$ Randomly sample a step from $\Omega_{step}$; 
 $\quad\mid$ Take the sampled step; 
 **end** |

---

## A.4 DATA PREPARATION

We used three datasets in the paper: Colored MNIST dataset (Kim et al., 2019), ModelNet40 dataset (Su et al., 2015) and NVGesture dataset (Molchanov et al., 2015), as illustrated in Figure 1.

For Colored MNIST and ModelNet40, we did not perform any extra data pre-processing steps on the original datasets.

For the NVGesture dataset, each video has a resolution of $240 \times 320$ and a duration of 80 frames from action starting to ending. There are three videos with unmatched starting indices between RGB and depth. We adopt the starting frame indice of RGB for all modalities. We randomly select 64 consecutive frames from the videos in the dataset and if the video has less than 64 frames, we zero-pad on both sides of it to obtain 64 frames. Frames are resized as $256 \times 256$ and are cropped into $224 \times 224$ as inputs (we use static cropping where we crop from the same location across times and modalities).

During training, we perform spatial augmentation on the video, including flipping and random cropping. During inference on validation or test set, we perform center cropping on the video.

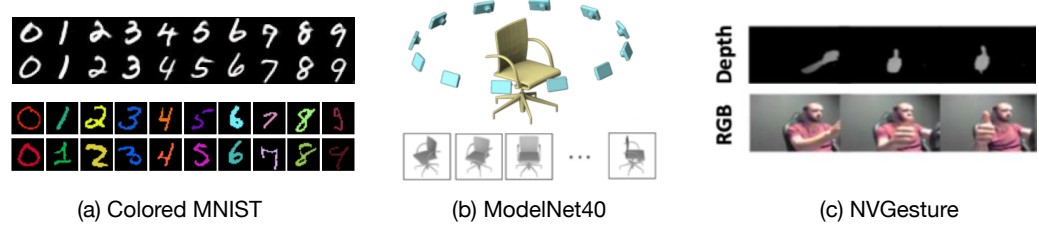

(a) Colored MNIST            (b) ModelNet40            (c) NVGesture

Figure 1: (a) The Colored MNIST dataset (Kim et al., 2019). We consider the monochromatic image, and the gray-scale image as the two input modalities (b) The ModelNet40 dataset (Su et al., 2015). The 2D representations are gray-scale images rendered from 12 different viewpoints of the object. (c)The NVGesture dataset (Molchanov et al., 2015). We use depth and RGB channels as the two modalities.

## A.5 MORE RESULTS

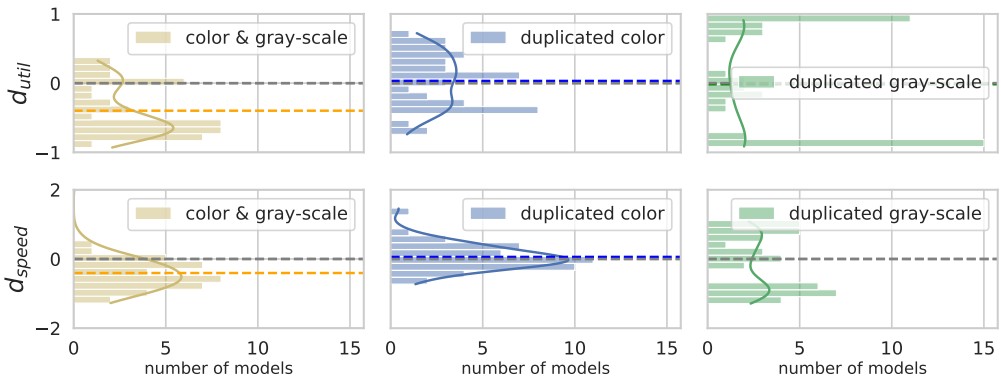

Figure 2: Histograms and estimated density functions of $d_{util}$ and $d_{speed}$ of models trained for colored-and-gray-MNIST, using monochromatic and gray-scale images as two modalities, using identical monochromatic images as two modalities and using identical gray-scale images as two modalities.

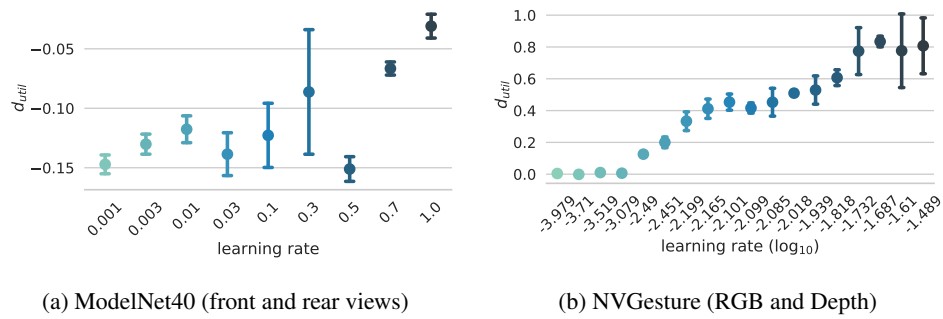

(a) ModelNet40 (front and rear views)      (b) NVGesture (RGB and Depth)

Figure 3: The imbalance in utilization, measured by $d_{util}$ for models trained using different learning rate. It appears that high learning rates can help with mitigating the imbalance in utilization between modalities.

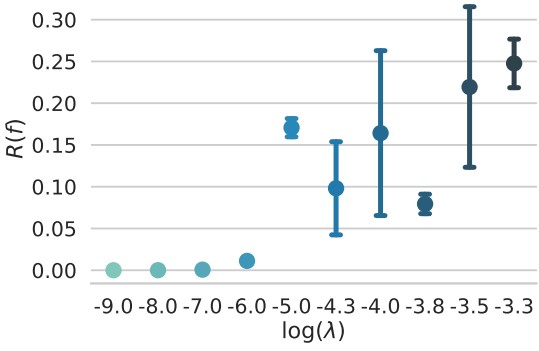

Figure 4: The mean and standard deviation of $R(f)$ for three model repetitions obtained by training the model with $\lambda$ as the weight on the L1 regularization. Generally, the larger the $\lambda$ is, the higher the $R(f)$ is, i.e., the sparser the model's parameters are.

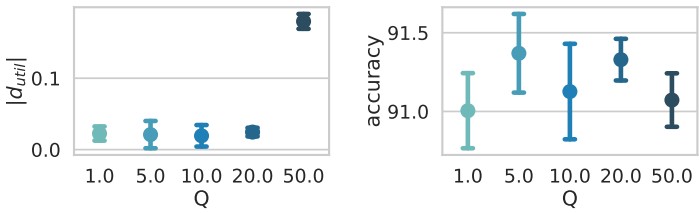

Figure 5: Models' behavior when using different values for the re-balancing window size $Q$ in the balanced multi-modal training algorithm. We use ModelNet40 (front and rear views) in the study. We fix the learning rate at 0.1 and the imbalance tolerance parameter at 0.01 while using $Q$ of 1, 5, 10, 20 and 50. We can effectively control the imbalance in conditional utilization except for $Q = 50$ (see $d_{util}$ shown in the left panel). According to the accuracy the model reaches, we choose to use $Q = 5$.

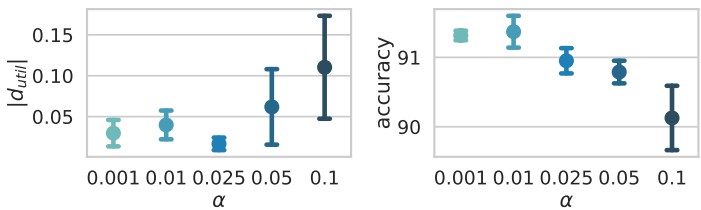

Figure 6: Models' behavior when using different values for the imbalance tolerance parameter $\alpha$ in the balanced multi-modal training algorithm. We use ModelNet40 (front and rear views) in the study and fix the learning rate at 0.1 and the re-balancing window size at 5. The values we use for $\alpha$ are ratios of $\mathbb{E}(\widehat{d}_{util})$ computed in the study in §5.2. Precisely, we use $0.01\mathbb{E}(\widehat{d}_{util})$, $0.1\mathbb{E}(\widehat{d}_{util})$, $0.25\mathbb{E}(\widehat{d}_{util})$, $0.5\mathbb{E}(\widehat{d}_{util})$, $\mathbb{E}(\widehat{d}_{util})$. Based on the pattern of $d_{util}$ shown in the left panel, using less than 0.25 of $\mathbb{E}(\widehat{d}_{util})$ gives desirable results. We choose to use $\alpha = 0.1\mathbb{E}(\widehat{d}_{util}) = 0.01$ according to the accuracy.

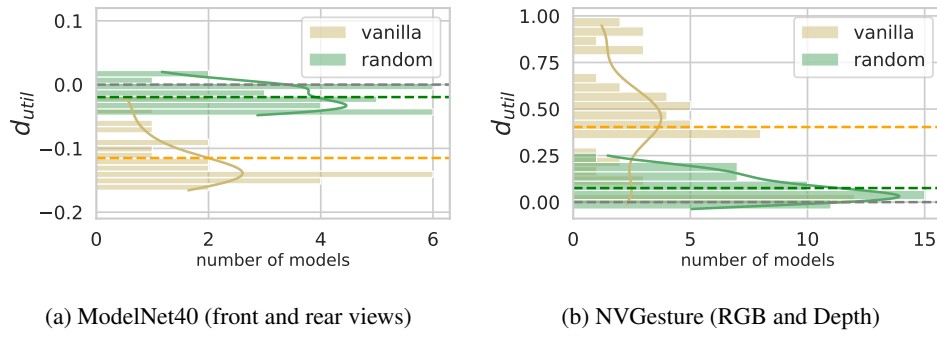

(a) ModelNet40 (front and rear views)    (b) NVGesture (RGB and Depth)

Figure 7: Histograms and estimated density functions of $d_{util}$ for random version of the balanced multi-modal learning process and the vanilla process. Both versions of the balanced multi-modal learning process are less greedy than the vanilla one.