# OpenReview forum: "Recognizing and overcoming the greedy nature of learning in multi-modal deep neural networks"
_ICLR.cc/2022/Conference — ICLR 2022 Submitted_

### Official Review · Reviewer_BHoa · 2021-10-29

**Correctness:** 3
**Technical Novelty And Significance:** 3
**Empirical Novelty And Significance:** 3
**Recommendation:** 5
**Confidence:** 4

**Main Review:**

1.	The topic of modal balance is interesting and has attracted much attention. The proposed method is direct and easy to follow.
2.	Experiments on several datasets verify the effectiveness.


weakness
1.	How to set the \alpha at the bottom of page five? More parameter analyses are expected.
2.	As introduced in the introduction and abstract, many related works about this topic have been proposed, e.g., “Rubi: Reducing unimodal biases in visual question answering”, “Removing bias in multi-modal classifiers: Regularization by maximizing functional entropies”. What are the improvements to the proposed methods? Why not compare these methods? The reviewer believes that the experiments are very lacking in comparison with the SOTA methods.



**Summary Of The Paper:**

This paper focuses on the multi-modal interaction problem, that multi-modal models tend to rely on just one modality while under-utilizing the other modalities. Since conditional utilization rate cannot be computed efficiently during training, they introduce an efficient proxy based on the pace at which a DNN learns from each modality, which we refer to as conditional learning speed. So they propose a training algorithm, balanced multi-modal learning, and demonstrate that it indeed addresses the issue of greedy learning. The proposed algorithm is found to improve the model’s generalization.

**Summary Of The Review:**

This paper focuses on the multi-modal interaction problem, that multi-modal models tend to rely on just one modality while under-utilizing the other modalities. Since conditional utilization rate cannot be computed efficiently during training, they introduce an efficient proxy based on the pace at which a DNN learns from each modality, which we refer to as conditional learning speed. So they propose a training algorithm, balanced multi-modal learning, and demonstrate that it indeed addresses the issue of greedy learning. The proposed algorithm is found to improve the model’s generalization. So I recommend weak accept.

---

> ### Author Response · Authors · 2021-11-15
> **Authors' response to Reviewer BHoa**
>
> Thank you for your comments and thoughtful review. We address some specific points below.
>
> > How to set the \alpha at the bottom of page five? More parameter analyses are expected.
>
> We set Q as 5 for all experiments considering the number of mini-batches per epoch. In fact, in a later study, we found that as long as Q is not dramatically larger, for example than 20, models’ behavior is relatively consistent in terms of generalization performance and modality-wise utilization.
>
> We chose $\alpha$ based on the span of $d_{util}$ observed for the vanilla system and also conducted a preliminary study on the model’s sensitivity to varying $alpha$.
> We will provide studies on the hyperparameters in the revised version.
>
> > As introduced in the introduction and abstract, many related works about this topic have been proposed, e.g., “Rubi: Reducing unimodal biases in visual question answering”, “Removing bias in multi-modal classifiers: Regularization by maximizing functional entropies”. What are the improvements to the proposed methods? Why not compare these methods?
>
> We focus on tasks using multiple vision modalities and intermediate fusion multimodal networks. It has not been explored before in the literature and most papers inspect similar issues but for visual question answering tasks, such as the two mentioned by the reviewer. In fact, in our preliminary exploration, we implemented RubiLoss [1] on NVGesture-scratch and we obtained an accuracy of $79.95 \pm 0.12$. It is similar to our guided algorithm with an accuracy of $ 80.22 \pm 0.73$. We are working on extending this experiment to other datasets we used in the paper and we will provide the results in the revised version.
>
> [1] Remi Cadene, Corentin Dancette, Hedi Ben-Younes, Matthieu Cord, and Devi Parikh. Rubi: Reducing unimodal biases in visual question answering. arXiv:1906.10169, 2019.

---

> > ### Comment · Reviewer_BHoa · 2021-11-26
> > **Response**
> >
> > Thank you for the responses. The author answered part of my questions, but only rough answers were given in the experimental part without detailed results and analysis.

---

### Official Review · Reviewer_PUm3 · 2021-11-02

**Correctness:** 2
**Technical Novelty And Significance:** 2
**Empirical Novelty And Significance:** 3
**Recommendation:** 3
**Confidence:** 4

**Main Review:**

Strength:
This paper focuses on a very interesting problem in multi-modal learning: the imbalanced training between modalities. Researches about this issue are meaningful and the idea of controlling the training process specific to modality is an intuitively promising way.

Weakness:
1.	It seems the experiments verifies the greedy learner hypothesis, but no proofs or theories are provided to support this hypothesis, either detailed analysis why the training speeds are different.
2.	The re-balancing step method, which is used to make multi-modal learning less greedy, is not well described and analysis in the full text. The author should have a well explanation that why using the average feature of previous samples can stop the training of the uni-modal branch.
3.	Lack comparation with other methods that solve imbalance problem, including [1]. The discussion in related work section is also inadequate. For example, related work [2-3] are not discussed.
4.	The author set different imbalance parameter for different dataset, but did not provided the ablation study for this crucial parameter. Does it impact the model performance greatly?
5.	All datasets only contain visual modality. More natural modalities, like audio and visual, should be included for better evaluation.

[1] Weiyao Wang, Du Tran, and Matt Feiszli. What makes train- 1055 ing multi-modal classification networks hard? In Proceedings of the IEEE/CVF Conference on Computer Vision and Pattern Recognition, pages 12695–12705, 2020.
[2] Thomas Winterbottom, Sarah Xiao, Alistair McLean, and Noura Al Moubayed. On modality bias in the tvqa dataset. BMVC, 2020.
[3] Ya Sun, Sijie Mai, and Haifeng Hu. Learning to balance the learning rates between various modalities via adaptive tracking factor. IEEE Signal Processing Letters, 28:1650– 1654, 2021.





**Summary Of The Paper:**

This paper hypothesizes that due to the greedy nature of deep learning, these models tend to rely on just one modality while under-utilizing the other modalities. The authors empirically observe this phenomenon on several dataset with a proposed metric. They propose an algorithm to achieve balanced training, which forcing the model to update only one of the uni-modal branches when the two uni-modal branches are imbalanced. This method is applied to three datasets and gain some progress.

**Summary Of The Review:**

This paper focuses on an interesting phenomenon in the multi-modal learning: the imbalance problem in the multi-modal training. A training algorithm, balanced multi-modal learning, is proposed to solve this problem. According to the provided experiment results, this method addresses the issue of greedy learning. However, the author did not provide solid theory analysis about not only the greedy learning problem but also the proposed method. Related works are also not well discussed and compared.

---

> ### Author Response · Authors · 2021-11-15
> **Authors' response to Reviewer PUm3 (1/2)**
>
> We thank the reviewer for the time to read our manuscript and provide us with your feedback. We would appreciate it if you can reconsider your score.
>
> Below we provide our responses to specific comments. We are happy to expand the discussion to clarify our work further.
>
> > It seems the experiments verifies the greedy learner hypothesis, but no proofs or theories are provided to support this hypothesis, either detailed analysis why the training speeds are different.
>
> It is hard to theoretically prove the proposed hypothesis but we appreciate your recognition of the value and quality of our designed experiments and our empirical results validating the greedy learner hypothesis.
>
> Regarding the difference in the training speeds, we consider it as a fact that has been observed in previous studies and we did not verify it additionally in this work. As mentioned in section 3.1, Wang et. al[1] compared the Overfitting-to-Generalization Ratio, and Wu et.al [2] observed differences in gradient norm of model parameters related to different modalities. Additionally, the imbalance in conditional learning speeds we observed in our study also provides evidence of the different training speeds.
>
> [1] Weiyao Wang, Du Tran, and Matt Feiszli. What makes training multi-modal classification networks hard? CVPR, 2020a.
> [2] Nan Wu, Stanisław Jastrzębski, Jungkyu Park, Linda Moy, Kyunghyun Cho, and Krzysztof J. Geras. Improving the ability of deep networks to use information from multiple views in breast cancer screening. MIDL, 2020.
>
> > The re-balancing step method, which is used to make multi-modal learning less greedy, is not well described and analysis in the full text.
>
> Given the length limitations, we did not manage to put the detailed explanation in the manuscript but in *section A.2 The Rebalancing Step* in the appendix. We will reshuffle the contents of the paper to explain the re-balancing step better the main part of the paper in the revision.
>
> > The author should have a well explanation that why using the average feature of previous samples can stop the training of the uni-modal branch.
>
> We believe that there is a misunderstanding of our proposed re-balancing step and we will try to clarify these points in the paper.
>
> The goal of the re-balancing step is to encourage the training of one uni-modal branch by limiting cross-modality information flow from the fusion modules. We remove sample-wise differences in the outputs from the fusion module before passing them to the uni-modal branch. Specifically, we use the average feature of previous training samples as the outputs. Since there is almost no mutual information between these features and the target, to reduce the loss, the model has to learn from the one input modality by updating the parameters in the uni-modal branch.  Another objective of using the average feature is to avoid distribution shift. This is motivated by the batch normalization technique.

---

> > ### Author Response · Authors · 2021-11-15
> > **Authors' response to Reviewer PUm3 (2/2)**
> >
> > > Lack comparation with other methods that solve imbalance problem, including [1].
> >
> > We acknowledge that we did not provide results of implementing methods proposed in other works in the datasets we considered. We consider the greedy learner hypothesis as the focal point of our work and hope to offer an alternative lightweight method to the toolbox in regularizing intermediate fused multi-modal networks. As far as we have seen, there are not many methods proposed before in the literature of this type of network in tasks of multiple vision modalities. Most papers inspect similar issues in visual question answering tasks. The proposed methods in [1] are designed for late-fusion networks. We did not experiment with methods in [1] but provided a discussion on it in the related work.
> >
> > In our preliminary exploration, we implemented RubiLoss [2] on NVGesture-scratch and we obtained an accuracy of $79.95 \pm0.12$. It is similar to our guided algorithm with an accuracy of $80.22 \pm 0.73$. We are working on extending this experiment to other datasets we used in the paper and we will provide the results in the revised version.
> >
> > [1] Weiyao Wang, Du Tran, and Matt Feiszli. What makes training multimodal classification networks hard? In Proceedings of the IEEE/CVF Conference on Computer Vision and Pattern Recognition, pages 12695–12705, 2020.
> >
> > [2] Remi Cadene, Corentin Dancette, Hedi Ben-Younes, Matthieu Cord, and Devi Parikh. Rubi: Reducing unimodal biases in visual question answering. arXiv:1906.10169, 2019.
> >
> > > The discussion in the related work section is also inadequate. For example, related work [2-3] are not discussed.
> > > [2] Thomas Winterbottom, Sarah Xiao, Alistair McLean, and Noura Al Moubayed. On modality bias in the tvqa dataset. BMVC, 2020.
> > > [3] Ya Sun, Sijie Mai, and Haifeng Hu. Learning to balance the learning rates between various modalities via adaptive tracking factor. IEEE Signal Processing Letters, 28:1650– 1654, 2021.
> >
> > Thank you for these references. We will discuss them in the revised version of the paper.
> >
> > > The author set different imbalance parameter for different dataset, but did not provided the ablation study for this crucial parameter. Does it impact the model performance greatly?
> >
> > Thank you for pointing out this piece of the missing discussion. We set Q as 5 for all experiments considering the number of mini-batches per epoch. In fact, in a later study, we found that as long as Q is not dramatically large, for example, larger than 20, the model behaves relatively consistent in terms of generalization and modality-wise utilization. We chose $\alpha$ based on the span of $d_{util}$ observed for the vanilla system and also conducted a study on the model’s sensitivity to varying $\alpha$.  We will provide more details and a study on the hyperparameters in the revised version, in order to assist in applying the proposed methods in other multimodal learning tasks. We hope the below information resolves your concern.
> >
> > > All datasets only contain visual modality. More natural modalities, like audio and visual, should be included for better evaluation.
> >
> > We agree that we focus on datasets with multiple visual modalities. We would like to highlight that the proposed hypothesis and methods are validated to be true and effective on all the datasets and tasks we experimented with. Along with the fact that we did not introduce any special pre-requisite on the modalities and the model architectures, it is reasonable to assume that the insights from our work are generalizable to other scenarios. If you have a specific dataset in mind for which you would like to see the results, we are happy to conduct an additional experiment.
> >
> > > the author did not provide solid theory analysis about not only the greedy learning problem but also the proposed method.
> >
> > Our paper indeed provides a mostly empirical explanation for the well-observed phenomenon that it is hard to train end-to-end multimodal learning systems to achieve satisfactory performance. We are pioneering the empirical analysis of this phenomenon. We would love to give a more theoretical explanation but this will come after these phenomena are understood empirically. If you have a specific kind of theory that you would like to see, we are open to an in-depth discussion.

---

> > > ### Comment · Reviewer_PUm3 · 2021-11-25
> > > **Response to authors**
> > >
> > > Thanks for the authors' efforts. After viewing the reviews and other rebuttals, I tend to keep my rating. The topic of this paper is interesting and worthy of exploration, but the method and analysis in this paper is not well supported, except some empirical validation on several visual-only dataset, and the method is only compared with some simple baseline. I think the author should further thoroughly analyze the proposed method not only in theories but also in experiments, especially on dataset with natural modality (such as some audio-visual dataset, like Kinetics-Sounds).

---

> ### Author Response · Authors · 2021-11-22
> **A follow-up with Review PUm3**
>
> Dear reviewer, we would like to know whether our responses below addressed your concern. If so, did we manage to convince you to increase your score? We have also provided the revision and we are happy to answer any further questions and comments.

---

### Official Review · Reviewer_B4j4 · 2021-11-03

**Correctness:** 3
**Technical Novelty And Significance:** 4
**Empirical Novelty And Significance:** 4
**Recommendation:** 8
**Confidence:** 4

**Main Review:**

1. The paper focuses on a relevant problem in multi-modal learning on how to utilize information from both modalities efficiently.
The strength of the paper is that they follow a structured and scientific way of identifying the problem, proposing a metric to measure the extent of the problem and perform experiments
to validate their hypothesis. They also show through experiments that the proposed training procedure leads to more efficient modality utilization and shows improved accuracy. The empirical results are strong.
2. Weakness: In a MTMM architecture, Theta_0 (red blocks in Fig 1) would also be receiving updates based on the modality m1 as information is being passed through the fusion blocks denoted by parameters (theta_0)'(green and red-green blocks).
In that case, I am not sure if definition 4.1 is a valid way to define the conditional learning speeds as both theta_0 and theta_0' would see the information flow from modality m1. It would have been valid if theta_0 did not se information from m1 at all (like the way conditional utilization rate was defined in 2.1). So, I am not clearly convinced with the statement in Section 4.1 that "If the greedy learner hypothesis is true, the conditional learning speed would serve as a proxy for the conditional utilization rate. dspeed(f;t) would predict dutil(f) at the end of training." Authors have tried to explain how to derive uni-modal outputs from MTMM in section A1 but it isn't clear to me
3. Re-balancing step is not clear as explained in Appendix A2. Lets say we want to balance the information flow from modality m0 in a MTMM architecture in Fig 1.
In that case, does it mean the top branch takes information from just m0 (i.e. the information flow from green and green-red blocks is stopped) to generate prediction y0_hat, while the bottom branch takes information from both m0 and m1 to generate the prediction y1_hat.

**Summary Of The Paper:**

The paper talks about efficient multi-modal training. They hypothesize that multi-modal training is greedy and perform experiments to empirically show the same.
They propose metrics such as conditional utilization rate and conditional learning speed which serves as a proxy for the former to measure the imbalance in utilization between modalities while training.
Based on conditional learning speed, they propose a multi-modal training algorithm that mitigates the imbalance in modality utilization.
Through experiments, the authors empirically show that it is true and their training paradigm performs better than vanilla multi-modal training.

**Summary Of The Review:**

This paper is easy to read and addresses the shortcomings of vanilla multi-modal training scenarios. Authors have done a good job of explaining the problem followed by laying out their hypotheses clearly. They have done experiments which enforces the hypotheses empirically. Authors should explain the re-balancing step (in section A2) and how to derive uni-modal predictions from multi-modal DNNs (in section A1) more clearly with consistent notations as used throughout the paper.

---

> ### Author Response · Authors · 2021-11-15
> **Authors' response to Reviewer B4j4**
>
> We appreciate the reviewer’s careful review and feedback. We acknowledge the lack of clarity in sections A1 and A2 in the appendix. We will improve it in the revision.
> We address specific points raised by the reviewer below. Please do not hesitate to post additional comments and questions. We are happy to address them.
>
> > I am not sure if definition 4.1 is a valid way to define the conditional learning speeds as both theta_0 and theta_0' would see the information flow from modality m1.
>
> > So, I am not clearly convinced with the statement in Section 4.1 that "If the greedy learner hypothesis is true, the conditional learning speed would serve as a proxy for the conditional utilization rate. $d_{speed}(f;t)$ would predict $d_{util}(f)$ at the end of training."
>
> **As shown in Figure 2, the defined conditional learning speed captures the relative learning pace between two modalities empirically given a modality-specific loss.** For example, compared to parameters in the uni-modal network $\theta_0$, the fusion parameters $\theta’_0$ handle information from $m_1$ more directly. When the modality-specific loss $l_0$ reduces faster by learning from $m_1$,  $\theta’_0$ gets updated remarkably. When $m_0$ helps more in reducing $l_0$, $\theta_0$ is then updated largely.  It can serve as a proxy for the conditional utilization which we can use to monitor the training on-the-fly, without introducing extra modifications to the model.
>
> We understand your concern about the conditional learning speed. We agree with you that both $m_0$ and $m_1$ contribute to the update of $\theta_0$ since information flows from the fusion blocks to the uni-modal branch. However, it is impractical to fully decompose the model’s parameters completed by modalities.
>
> > Authors have tried to explain how to derive uni-modal outputs from MTMM in section A1 but it isn't clear to me.
>
> We are sorry about the typos in section A1. We shall change $f^\prime(x_{m_0})$ to  $f^\prime_0(x_{m_0})$ and $f^\prime(x_{m_1})$ to $f^\prime_0(x_{m_1})$.
>
> Specifically, to derive uni-modal outputs from MTMM, such as $f^\prime_0(x_{m_0})$, we pass the constant for all samples as input to the fusion module so that f’_0 is not aware of the other modality. To avoid distribution shift, we compute the constant by averaging inputs from $m_1$ over samples in the training set.
>
> > Re-balancing step is not clear as explained in Appendix A2. Lets say we want to balance the information flow from modality m0 in a MTMM architecture in Fig 1. In that case, does it mean the top branch takes information from just m0 (i.e. the information flow from green and green-red blocks is stopped) to generate prediction y0_hat, while the bottom branch takes information from both m0 and m1 to generate the prediction y1_hat.
>
> **Yes, your interpretation of the re-balancing step is correct.** As you assumed, if the algorithm detects that learning from $m_0$ requires acceleration, then we will limit the information from $m_1$ to flow to the uni-modal network (the top branch) through the corresponding fusion module (the green and green-red blocks). The other part of the networks will be running as normal.

---

### Official Review · Reviewer_ndMY · 2021-11-04

**Correctness:** 2
**Technical Novelty And Significance:** 2
**Empirical Novelty And Significance:** 2
**Recommendation:** 3
**Confidence:** 4

**Main Review:**

* Strengths
  - The greedy behaviors among modalities are frequently observed in practical learning process of multi-modal DNNs. The research topic of this paper is interesting and worthy of investigation.
  - The paper is easy to follow.

* Weaknesses
  - The main claims related to worse generalization performance from the greedy nature is not so clearly validated with experiments. It is critical issue since it is connected with the reasons of necessity of the proposed methods. Table 1 shows that the experimental results between *vanilla* and *guided* are not statistically significantly different in ModelNet40, NVGesture-scratch and NVGesture-pretrained. Even though those of Colored-and-gray-MNIST dataset seem to demonstrate big differences, I think the dataset is intentionally designed to be biased with color and labels and not compatible with standard multi-modal setting. So, only the results with Colored-and-gray-MNIST dataset are not enough to support the main claims.
   - The other claims on are also not so clear to accept. For Section 5.2, if the values of | d_util | are high, it seems to mean that there are differences of conditional utilization rates, not confirm that the standard multi-modal learning process encourages the model to rely on one modality and ignore the other one. In other words, I think the experimental design and their evidences are weak to validate their claims.


* Minors
  - Section 5.3 in page 8, obverse -> observe


**Summary Of The Paper:**

This paper investigates the modality greediness of learning in multi-modal deep neural networks (DNNs). The authors hypothesize that multi-modal DNNs learn to rely on one of input modalities (it could learn faster), which is expressed as a “greedy" behavior.
The greediness is quantified with the difference between conditional utilization rate between modalities.
They claim the followings: (1) conditional utilization rate can be surrogated with conditional learning speed, (2) the strong regularization in the training process encourages the greediness, and (3) the greedy behavior often lowers generalization performances.
As a remedy of performance decreasing phenomena, they propose balanced multi-modal learning methods, which control the training speeds of all of modalities to be within certain criterion via giving more steps considering the log-ratio of sum of the magnitudes of gradients of loss functions between of fusion model parameters and of unimodal model parameters.
The paper provides experimental results to validate their claims and the effectiveness of the proposed methods.


**Summary Of The Review:**

The research topic is interesting and it seems that the main answer points of the research questions are appropriate.
However, main claims in the paper are not so clearly validated with respect to both of theories and experiments.
I think that the authors should reconsider whether the differences on conditional utilization rate and those of conditional learning speed really affect the generalization performances, or they should try to establish other strategies to find evidences to support main claims.
From the reason, I recommend this paper as reject.

---

> ### Author Response · Authors · 2021-11-15
> **Authors' response to Reviewer ndMY**
>
> Thank you again for your efforts in reviewing the paper. We appreciate any feedback and we are happy to incorporate your suggestions in the revision.  We give some clarifications below. We are happy to expand the discussion further.
>
> > Table 1 shows that the experimental results between vanilla and guided are not statistically significantly different in ModelNet40, NVGesture-scratch and NVGesture-pretrained.
>
> Given the multi-view property of both ModelNet40 and NVGesture, it is non-trivial to gain 0.1 in accuracy since we did not introduce any changes to the models' architecture. More importantly, we see that the guided algorithm addresses the issue of greedy learning behavior as shown in Figure 4.
>
> > Even though those of Colored-and-gray-MNIST dataset seem to demonstrate big differences, I think the dataset is intentionally designed to be biased with color and labels and not compatible with standard multi-modal setting.
>
> There are different multimodal learning tasks, and they are on a broad spectrum. This task is on the far end of the spectrum where phenomena associated with multi-modal learning are sharper. Therefore our method exhibits a notable performance gain.
>
> > For Section 5.2, if the values of | d_util | are high, it seems to mean that there are differences oconditional utilization rates, not confirm that the standard multi-modal learning process encourages the model to rely on one modality and ignore the other one.
>
> **High |$d_{util}$|, especially being higher than 0.5, does indicate that the model relies on one modality and ignores the other one when making predictions.** Assuming $u(m_0 | m_1) = \big(A(f_1)−A(f^\prime_1)\big)  \/A(f_1) \sim 0$, we see that the accuracy of $y_1$ doesn't change much whenever we let it access modality $m_0$ or not, as long as it can use $m_1$. Now  we have $u(m_1 |m_0)\sim 1$, it means that $m_1$ has a strong impact on the accuracy of $y_0$, and it is not enough for the model to use merely $m_0$ to make $y_0$ accurate. Above all, we have $m_1$ impact significantly the accuracy of $y_0$ and $y_1$ and $m_0$ contributes marginally to $y_0$ and $y_1$, when $ |d_{util}| =  |u(m_1 |m_0) - u(m_0 |m_1)|$ is high.
>
> > Differences on conditional utilization rate and those of conditional learning speed really affect the generalization performances.
>
> **Differences observed in conditional utilization rate and conditional learning speed do explain the model’s imperfect generalization performance.** Our study is based on this agreement that the model’s ability to comprehend all modalities is the foundation of its generalization. Using conditional utilization rates, we observe explicitly that models, trained by the vanilla multimodal learning algorithm, rely on one modality while under-fitting the others. We show that they achieve better generalization if we can help them to overcome such greediness.

---

> > ### Comment · Reviewer_ndMY · 2021-11-25
> > **Response to the authors**
> >
> > Thank you for the responses.
> >
> > I've read all of the other reviews and their responses including mine.
> > While the revised manuscript has been improved with respect to experiments, writing and readability, the supports of main claims still seems not to be so strong and convincing.
> > For example, Table 1 is augmented with the results of RUBi.
> > However, some other numbers and bolded ones in Table 1 are changed in ModelNet40 and NVGesture-scratch without any changes on experimental settings in the manuscript.
> > Which one is correct between the first version and the revised one?
> >
> > Also, I still think that it needs more concrete and connected chains verified among the claims.
> > Even though we observe some positive cases in real worlds, we should validate postulated arguments to be true.
> > Even though this manuscript provides some promising tendency with experimental results, I believe that they are not enough to be validation.
> > I tend to keep my score.

---

> ### Author Response · Authors · 2021-11-22
> **A follow-up with reviewer ndMY**
>
> Dear reviewer, did we manage to convince you to increase your score, with the responses below and the revision? We are happy to answer any further questions and comments.

---

### Author Response · Authors · 2021-11-18
**Rebuttal Revision**

We would like to thank all reviewers for their constructive suggestions. We revised the paper accordingly:

*  We fixed typos pointed out by reviewer ndMY and B4j4 in the manuscript and the appendix.
*  We augmented related works with papers the reviewer PUm3 mentioned.
*  We provided studies on the model's sensitivity to the hyperparameters introduced in the proposed balanced multi-modal learning algorithm. We present the results in the appendix in **Figure 5** about the re-balancing window size, $Q$, and **Figure 6** about the imbalance tolerance parameter, $\alpha$.
* We applied RUBi learning strategy[1] on the four tasks considered in **Section 5.4: Improved generalization performance**. We updated Table 1 accordingly with the results obtained. We did not find RUBi learning strategy to be effective. Our proposed algorithm performs better than all other three methods.
* We revised and polished the manuscript to improve readability.

We hope the above updates resolve concerns the reviewers had and enhance the quality of our work. We are happy to discuss any of the changes in detail.

[1]  Remi Cadene, Corentin Dancette, Hedi Ben-Younes, Matthieu Cord, and Devi Parikh. Rubi: Reducing unimodal biases in visual question answering. arXiv:1906.10169, 2019.

---

### Decision · Program_Chairs · 2022-01-20

**Decision:**

Reject

**Comment:**

PAPER: This paper presents analysis of cross-modal interactions in multimodal models and propose a method to help balance the multimodal learning process. The cross-modal analysis is based on measures related to conditional utilization rate and the proposed approach is related to conditional learning speed.
DISCUSSION: The reviewers showed support for this line of research, as a way to better understand the learning process for multimodal models. The discussion allowed to identify points that needed to be clarify and concerns about the experimental results. The authors addressed many of these issues in their response. All reviewers took the time to read these responses as well as other reviews. There are many reviewers who are still expressing concerns with the experimental results.
SUMMARY: This is an important line of research, and the authors should continue this research endeavor. While the paper presents some interesting research hypotheses about multimodal learning, it seems that more experiments are needed to properly address these hypotheses. In its current form, the paper may not yet be ready for publication.